# Effect of Winemaking on Phenolic Compounds and Antioxidant Activities of Msalais Wine

**DOI:** 10.3390/molecules28031250

**Published:** 2023-01-27

**Authors:** Xiaojie Hou, Shenghuizi Chen, Yunfeng Pu, Tingting Wang, Heng Xu, Hu Li, Peng Ma, Xujie Hou

**Affiliations:** 1College of Food Science and Engineering, Tarim University, Alar 843300, China; 2Production & Construction Group Key Laboratory of Special Agricultural Products Further Processing in Southern Xinjiang, Alar 843300, China; 3College of Food Science and Engineering, Xinjiang Institute of Technology, Akesu 843000, China; 4Regional Center for Disease Prevention and Control, Akesu 843000, China

**Keywords:** Msalais wine, LC-MS/QTOF, phenolic compounds, antioxidant activity, winemaking

## Abstract

Msalais wine (MW) is a popular traditional wine with the cultural characteristics of a specific Chinese ethnic group. In this study, phenolic profiles and antioxidant characteristics were identified using chromatographic analysis. A total of thirty-eight compounds, including eight furans, eleven phenolic acids, fourteen flavonoids, and five others, were identified via LC-MS/QTOF. It was found that catechin is the most abundant phenolic compound in MW, followed by epicatechin, gallic acid, caffeic acid, rutin, and p-coumaric acid. Winemaking had a significant influence on the levels of phenols and antioxidant activity. Condensed juice (CJ) displayed the highest phenol and antioxidant activity levels, while the levels were significantly decreased during the fermentation process and gradually stabilized thereafter during the aging process. A correlation analysis between the polyphenols in Msalais and their antioxidant capacity was performed to determine which molecules contributed more to the antioxidant capacity in a complex mixture of polyphenols. All of the phenolic compounds, except ferulic acid, showed good correlation with DPPH, ABTS, and CUPRAC. Among them, resveratrol had the strongest antioxidant capacity, although its concentration was very low. Catechin also had a strong antioxidant capacity, which was positively correlated with its concentration. This indicates that the antioxidant activity of Msalais is related to the number, type, and structure of polyphenols.

## 1. Introduction

Traditional foods, which represent the heritage and dietary intake of a local geographical area, are key components for that particular region or country [1]. Msalais is a traditional wine mainly produced in Xinjiang, China, and it is produced by the Uygur ethnic group [2]. The production technique and quality of Msalais wine differs from conventional wine (Figure 1) [3]. It can be produced using the traditional method, but this is a time-consuming process. Condensed grape (*Vitis vinifera* Hetianhong) juice mainly prepared using an open hot cooking method from local cultivars [3]. After condensation, about two months of fermentation and 1 to 3 years of aging are needed at ambient temperatures [2]. Due to its strong regional characteristics and unique taste profile, in 2006, traditional Msalais wine was certified as a geographical indication protection product (Registration No. 5717691) by the national administration of quality supervision, inspection, and quarantine.

In recent years, Msalais has received increasing scholarly attention, but at this stage, there are few studies regarding Msalais in the international arena. Most of the current studies regarding Msalais have focused on the screening of brewer’s yeast [4,5,6] and the identification of volatile components [3,7,8]. Zhu [4,6] examined and analyzed the technical and sensory characteristics of 10 native strains identified from a collection of 436 native brewing strains, domesticated various strains suitable for the industrial production of Msalais, and evaluated the brewing characteristics of native brewing yeast strains related to Msalais on a pilot scale. Meanwhile, in response to defects in the traditional Msalais process, such as the long fermentation time, difficulty in controlling the fermentation process, and the lack of product stability, some scholars conducted the single-strain culture of 175 strains of Msalais yeast. They initially screened out two which were suspected to be brewer’s yeast and 20 non-brewer’s yeast strains with excellent aroma characteristics; then, they assessed different strains, different ratios of compounding, and different time sequence inoculations; verified the microfermentation; and obtained groups of bacteria, N8 (Saccharomyces cerevisiae) and F10 (Pichia kudriavzevii), in a 2:1 ratio [5]. In terms of aroma, sensory tests and gas chromatography–mass spectrometry analyses were first used to determine the changes in the aromatic compounds in Msalais during the production process [3], and then, a partial least squares regression analysis based on quantitative sensory description and odor activity values (OAVs), aroma extraction dilution analysis, and aroma recombination and omission tests were performed to outline the unique aroma characteristics of traditional Msalais [7]. The process of furfuryl alcohol production in Msalais, which produces the very characteristic caramel odor, was also investigated during the brewing process [8]. Among other aspects, as Msalais is fermented using grape juice concentrate, Zhang [9] investigated the evolution of non-enzymatic browning during Msalais production. Twelve physicochemical parameters in fifteen traditional Msalais wine samples were comprehensively analyzed by Zhu [2] using data mining methods to understand the physicochemical properties and homogeneity among Msalais samples and the interactions between them. It could be seen that there is a gap in the study of phenolic chemistry and antioxidant activity during the production of Msalais.

Phenols are a family of bioactive compounds found in grapes; therefore, derivatives such as wine and juices are rich in these bioactive compounds. These compounds improve organoleptic properties and chemical stability, and are associated with various health benefits [10,11]. Generally, a variety of bioactive compounds, including polyphenols, are changed during the brewing process. It has been reported in a range of studies that wine is rich in antioxidants, including polyphenols, which can terminate the free-radical chain reaction and chelate with metal ions [12,13]. It also been reported that the antioxidant capacity of foods is highly correlated with their phenols [14]. Zhu [2] reported that the total phenolic content in Msalais wines ranged from 16.12 to 58.75 mg/L [2]. Phenolic compounds and antioxidant properties in wine come from grape berries during the fermentation, aging, and storage of wine. The evaluation of phenolic and antioxidant profiling is very complex and intricate, and limited information is available regarding the phenolic composition and antioxidant potential. To this end, for the high-quality production of this economically important product, the concentration of phenolic and antioxidant profiling in Msalais wine at different stages of processing must be studied urgently. However, the changes in phenolic compounds and antioxidant characteristics during the brewing process of Msalais are not fully known to processors [3].

Therefore, in the present study, our aim was to provide the accurate and comprehensive identification of the phenolic constituents found in Msalais wine using high-performance liquid chromatography (HPLC) coupled with the quadrupole time of flight mass spectrometry (LC-MS/QTOF). The main focus of this study was to investigate the changes in the levels of phenols and antioxidant characteristics during the brewing process for Msalais wine.

## 2. Results and Discussion

### 2.1. Compounds Identified in Msalais Wine

Msalais wine was analyzed using high-resolution mass spectrometry, and the results are presented in Section 3.2. In total, 38 compounds were found, including 8 furans, 11 phenolic acids, 14 flavonoids, and 5 others, which were identified using LC-MS/QTOF in the negative ion mode (Table 1 and Appendix A). All of the compounds were tentatively identified by comparing their MS spectral data with those generated by the PeakView 1.0 software to produce empirical formulae, the chemical structures and names of which were obtained from the database of Chemspider, and which were further confirmed based on *m*/*z* values compared with authentic standards and literature reports.

### 2.2. Furans

In this study, a total of eight furan compounds were detected in Msalais wine (MW). According to the MS fragment behaviors, the compounds of C1, C2, C4-C7, C13, and C15 matched well with those in the databases, and they were tentatively identified as (3,4-dihydroxy-5-oxotetrahydro-2-furanyl)(hydroxy)-acetic acid, L-erytho-pent-1-enofuranose, 3-hydroxy-2-butanyl 6-O-[3,4-dihydroxy-4-(hydroxymethyl)tetrahydro-2-furanyl]-beta-D-glucopyranoside, 5-(hydroxymethyl)-2-furaldehyde, 3-hydroxyl-5-[1,2,3,4-tetrahydroxy butyl]-2(H)-furanone,[4-(methoxycarbonyl)-5-methyl-2furyl]methyl-2,3-dyhydro-1,4-benzodioxine-2-carboxylate, 2-isopropyl-5-methylphenyl 6-O-[3,4-dihydroxy-4-(hydroxymethyl)tetrahydro-2-furanyl]-β-D-glucopyranoside, and 1-(5-hydroxy-1-benzofuran-3-yl)thenone.

Furans are not commonly found in raw agricultural commodities, but they can be formed in heated, irradiated, or ultraviolet-treated food, such as canned fruits and vegetable products [15]. Msalais is a traditional wine, produced by the spontaneous fermentation of boiled grape juice, meaning that furans are inevitably produced during Msalais production. Additionally, Zhu [7] analyzed furaneol in Msalais wines and suggested that furaneol could be used as an important indicator of wine quality.

### 2.3. Phenolic Acids

In this study, eleven phenolic acids were identified in Msalais wine using the MS and literature data (Table 1). The names of the acids were determined, including garlic acid (*m*/*z* 169, C3), protocatechuic acid (*m*/*z* 153, C10), p-hydroxybenzoic acid (*m*/*z* 137, C14), ferulic acid (*m*/*z* 193, C19), caffeic acid (*m*/*z* 179, C20), p-hydroxycinnamic acid (*m*/*z* 163, C25), and ellagic acid (*m*/*z* 301, C28), which were elucidated by comparison with their ion fragmentation patterns and the comparison of the retention times with those of the authentic standards. Compound C11 (*m*/*z* 153.0564) was tentatively identified as gentistic acid. Its fragmentation pattern was consistent with the earlier report [16]. The compounds of C12 produced an [M-H]- ion at *m*/*z* 311.0396, which was assigned to caftaric acid. Its MS2 fragment ions at *m*/*z* 179 and *m*/*z* 135, which corresponded to [M-H-C4H4O5]- and [M-H-C4H4O5-CO2]–, respectively, were consistent with the report by Samoticha et al. [17]. The compounds of C17 produced an [M-H]- ion at *m*/*z* 295.0456 and MS2 fragment ions at *m*/*z* 163, which were also consistent with the report by Barnaba et al. [18].

### 2.4. Flavonoids

Flavonoids are a large family of phytochemicals with a general chemical structure of a 15-carbon skeleton, which consists of two phenyl rings (A and B) and a heterocyclic ring (C), and they have biological and pharmacological activities which include antioxidative, anti-inflammatory, antiviral, and antidiabetic effects [19]. Compounds C16, C21, C27, C29-C33, and C35-C37 were identified as flavonoids, and as well as those compounds, two flavanols, six quercetin derivatives, three Kaempferol derivatives, and three other flavones were found.

Catechin (*m*/*z* 289.0712, C16), epicatechin (*m*/*z* 289.0743, C21), and some other typical polyphenols and grape products were also detected in Msalais wine, and they were confirmed by comparison with authentic standards. These polyphenols were similar, as reported in earlier studies regarding wine [20]. Their fragmentations at *m*/*z* 289, 245, 203, and 151, which resulted from the loss of formic acid, are typical of epicatechin isomers, which directly correlates with the reports of studies in [21]. Additionally, this is consistent with the fact that the retention time of catechin was consistently less than that of epicatechin during chromatographic analysis.

Quercetin was identified in the MW (*m*/*z* 301.0354, C37) by comparison with an available standard. The mass spectra showed the ions corresponded to retro-Diels Alder fission with a cleavage of the heterocyclic ring (*m*/*z* 151) and a fragmentation by which a reopening and reclosing of the heterocyclic ring, along with a loss of a carbon in this ring, was observed (*m*/*z* 179) [22]. Quercetin derivatives were also identified in the MW as eriocitrin (*m*/*z* 595.1743, C22), including rutin (*m*/*z* 609.1453, C27), quercetin 3-O-glucuronide (*m*/*z* 477.0660, C20), quercetin-3-O-glucoside (*m*/*z* 463.0949, C29), and quercetin-O-rhamnoside (*m*/*z* 447.0908, C33). They shared the same fragment ions at *m*/*z* 301 and 300 in a full-scan mode in the triple quadrupole system, which are usually considered as key fragment ions of quercetin derivatives [16,23].

In this study, some compounds such as C31 (*m*/*z* 593.1511) and C36 (*m*/*z* 431.1013) had fragment ions at *m*/*z* 284 and 285, which indicated a heterolytic cleavage of the sugar moiety attributed to Kaempferol. The fragmentation patterns of these two compounds were in agreement with those presented in previous reports [24]. Therefore, compounds C31, C36, and C38 were tentatively identified as kaempferol-3-O-rutinoside, kaempferol-7-O-rhamnose, and Kaempferol, respectively, by analyzing and matching them with those obtained from the database of Chemspider. Moreover, kaempferol was further confirmed by comparison with the retention time and fragment ions of a standard chemical.

In addition to flavanols, quercetin derivatives and kaempferol derivatives are described in 3.1.2 above. Moreover, compound C35 (*m*/*z* 436.1224) was identified as phloridzin, which has a parent ion [M−H]- at *m*/*z* 435 and a fragment ion [M−H− Glu]- at *m*/*z* 273, which was further confirmed by comparison with the authentic standard. In this study, the C32 compound also had fragment ions at *m*/*z* 315 and 314, just 15 more (CH3-) than the characteristic ions of quercetin derivatives. It could be deduced that C32 belonged to flavonoids, and the formula and fragment ions could match those of ramnazin-3-O-rutinoside. C22 (*m*/*z* 449.1078) had fragment ions at *m*/*z* 287, which indicated a heterolytic cleavage of the sugar moiety attributed to eriodictyol, and it was tentatively identified as eriodictyol-7-O-glucopyranoside.

### 2.5. Others

Resveratrol has been reported as an important functional component in wine and grape products [20,25,26]. Compound C34 (*m*/*z* 227.0712) was identified as resveratrol and further confirmed by comparison with the authentic standard. Compounds such as C8, C9, and C23 with the molecular formulae of C20H30O13, C14H20O8, and C18H24O9, and molecular ions [M−H]− at *m*/*z* 477.1592, 315.1081, and 383.1382, respectively, were identified as hydroxyl benzenes by analyzing and matching them with those obtained from the database of Chemspider.

According to the present LC-MS/MS results, most phytochemicals in MW were reported in grape and its products. Flavanols, anthocyanins, and phenolic acids were reported as the major phenolic compounds in wine and grape juice [27]. Among them, anthocyanins are the main compounds responsible for the color of wine and grape juices, but anthocyanins were not identified in Msalais wine. *Vitis vinifera Hetianhong* belongs to a light-colored variety (Appendix A), which hardly contains anthocyanins. Grape juice should be concentrated before spontaneous fermentation, and some heat-sensitive components might be degraded or destroyed during the boiling process, which might be responsible for the absence of anthocyanins in MW.

### 2.6. Phenolic Compounds

The determined concentrations of phenolic compounds in grape juice (GJ), condensed juice (CJ), Msalais wine (MW), and aged Msalais wine (AW) are displayed in Table 2. The results show that about 25 phenolic compounds were identified in Msalais wine (MW) using LC-MS/MS. Moreover, the phenolic profiles of GJ, CJ, MW, and AW were analyzed, and nine different phenolic compounds were found, namely, four phenolic acids, two flavonols, two flavonoids, and one stilbene. The dominant phenolic compounds (Table 2) were found to be in condensation juice (CJ), which might have been due to the tendency of the system’s volume to reduce during the concentration process. These values significantly fluctuated within the fermentation process due to the hydrolysis and oxidation of polyphenols, whereas they gradually became stabilized during the aging process [28]. In this study, the catechin compound was found to be dominant, and this agreement matched with studies in which it was reported that the most abundant flavan-3-ol monomer found in grape juice was in the form of catechin [29].

During the concentration process, it was shown that the changes in the phenolic compounds were similar to those reported in previous studies, in which both the quercetin-3-O-rhamnoside and rutin levels in fruit decreased (8–92%) after hot-air/oven drying at 47–90 °C for 7–36 h [30], while the catechin and epicatechin levels increased approximately 0.2–10-fold. The reason behind these changes is that drying can break down cell walls and/or release them from sequestration, which may also result in compounds having a higher level of extractability from the samples [31]. The results obtained during the fermentation and aging processes also matched with reported studies in which wine was produced from various varieties of grape [32,33].

The level of catechin content increased from 89.65 mg/L to 272.92 mg/L in CJ, which was due to the increasing concentration, but it decreased by up to 201.03 mg/L in MW and also gradually decreased during the aging process. Epicatechin and catechin are the most common flavan-3-ol monomers [34], and they both decreased during the fermentation and aging stages because wine is a very complex medium. They underwent oxidation, polymerization, and pigmentation chemical reactions during the fermentation and aging stages [35,36].

Hydroxybenzoic acid is necessary for the synthesis of other compounds during the growth and development of grape berries. Gallic acid is considered the most abundant benzoic acid and an important phenolic compound because it is the precursor of all hydrolyzable tannins and is a condensed tannin [37]. The highest levels of gallic acid content were observed in CJ, MW, and AW. After the fermentation of CJ to obtain MW, the level of gallic acid increased from 26.48 mg/L to 38.01 mg/L, and subsequently, after being aged, the level of gallic acid continued to increase to 41.60 mg/L. This may have been due to the fact that the basic source of gallic acid is the fruit itself, and it is also formed via the hydrolysis of hydrolysable condensed tannins [38].

Caffeic acid, p-coumaric acid, and ferulic acid are all hydroxycinnamic acids with C6-C3 structures and are phenolic substances with high content levels in grapes and wines. They are susceptible to oxidation reactions that affect the color of wine and they are also precursors to some volatile components [39]. The highest level of p-coumaric acid was found in GJ. The highest level of caffeic acid was found in CJ, MW, and AW. The levels of caffeic acid and ferulic acid did not change much from fermentation to aging. Caffeic acid is important in fruit wines, as it increases the level of acylated anthocyanins, increases color stability through direct interactions between anthocyanins and free phenolic acids, and improves the stability of fruit wines during storage [40]. The level of p-coumaric acid decreased continuously from 13.03 mg/L to 7.38 mg/L during the fermentation and aging processes.

The level of resveratrol decreased by about 39% during the fermentation phase, which may have been due to the metabolic and hydrolytic reactions of the yeast [41]. The content of resveratrol in AW was slightly higher than that in MW, which could have been caused by evaporation during the fermentation process [41]. The most important flavonol compound in wine is quercetin, which increases the bitterness of wine and interacts with anthocyanins to stabilize its color [42]. The levels of rutin were relatively high in grape juice, whereas the change in rutin levels was obvious when compared to other compounds [38].

The level of rutin was found to increase by up to 53.45% during the concentration process, which continued to increase during the fermentation and aging stages. The quercetin content did not change greatly during the aging stage; however, the quercetin content in CJ was 1.98 mg/L, and the quercetin content increased by about 50% through fermentation in MW. Therefore, during alcoholic fermentation, because the solubility of quercetin in water is lower than that of anthocyanins, an increase in the amount of ethanol can promote the extraction of quercetin [32].

### 2.7. Antioxidant Activities

It is well known that reactive oxidants are generated in the human body, which act as poison in different ways and induce many chronic diseases such as mutation, cancer, atherosclerosis, and cardiovascular diseases [43]. Antioxidant capacity assays are necessary for the provision of regulatory standards regarding food quality and health claims [44]. Many studies have shown that phenolic compounds play an important role in the antioxidant activity of wine [45]. The effects of the different stages of the Msalais winemaking process on the antioxidant activities of the samples are shown in Figure 2. Despite the different mechanisms of the antioxidant assays, the results of the DPPH, ABTS, and CUPRAC assays revealed quite similar trends. The antioxidant activity increased dramatically during the concentration process, which was due to the simultaneous dramatic increase in the concentration of phenolic compounds during the concentration process. During the brewing process, a considerable number of phenolic substances are degraded or oxidized, other chemical reactions occur, and the phenolic substances in wine undergo significant changes, meaning the antioxidant activity level is significantly reduced during fermentation (*p* < 0.05) [45]. The antioxidant activity of Msalais stabilized during aging, a finding that could be attributed to the lack of anthocyanins in the molecular weight [14,46].

Pearson correlation coefficients between the individual phenolic compounds and the antioxidant activity of samples at different Msalais winemaking stages are shown in Table 3, which shows that all the phenolic compounds had a positive value and a good correlation with DPPH, ABTS, and CUPRAC, except ferulic acid (r > 0.5; *p* < 0.05). Additionally, statistically significant correlations (*p* < 0.05) were observed between phenolic compounds and the antioxidant activity. In the CUPRAC assay, the antioxidant capacity of resveratrol (resveratrol to CUPRAC: r = 0.978, *p* < 0.05) and catechin (catechin to CUPRAC: r = 0.978, *p* < 0.05) was highly significant; in the ABTS assay, the strongest antioxidant capacity was found in resveratrol (resveratrol to ABTS: r = 0.979, *p* < 0.05) and epicatechin (catechin to ABTS: r = 0.979, *p* < 0.05); resveratrol to ABTS: r = 0.979, *p* < 0.05) was followed by catechin (catechin to ABTS: r = 0.978, *p* < 0.05) and epicatechin (epicatechin to ABTS: r = 0.966, *p* < 0.05); in the DPPH assay, resveratrol (resveratrol to ABTS: r = 0.966, *p* < 0.05) was the most powerful antioxidant. It was shown that resveratrol (resveratrol to DPPH: r = 0.990, *p* < 0.05) had the strongest antioxidant capacity.

Catechin and epicatechin have strong antioxidant activity levels, which are strongly related to their chemical structural characteristics, probably because their b-ring has a catechol-type structure, which acts as a reducing agent [47]. Catechin, as the most concentrated phenolic compound in Msalais, has a high level of antioxidant activity, which fully indicates that the concentration of catechin is positively correlated with antioxidant activity. Viljanen, K derived the same results [48].

In Msalais, we found that the concentration of resveratrol was not high, but it had the strongest antioxidant activity, which is consistent with the results presented in studies regarding resveratrol with extremely strong antioxidant, antibacterial, and antifungal effects and cardioprotective, neuroprotective, and anticancer effects studied in the relevant literature [45,49]. This also indicates that the antioxidant activity depends not only on the number of polyphenols, but also on the type and structural characteristics of the polyphenols [50,51].

## 3. Materials and Methods

### 3.1. Chemicals

Gallic acid, catechin, epicatechin, chlorogenic acid, p-coumaric acid, caffeic acid, ferulic acid, coumarin, rutin, resveratrol, quercetin, 2,2′-azino-di-(3-ethyl-benzothialozine-sulphonic acid) (ABTS), 1,1-diphenyl-2-picrylhydrazyl (DPPH), and 6-hydroxy-2,5,7,8-tetramethylchroman-2-carbox-ylic acid (Trolox) were purchased from Sigma-Aldrich (St. Louis, MO, USA). Methanol and acetonitrile were purchased from Thermo Fisher Scientific, and other chemicals were obtained from Sinopharm Chemical Reagent Co. (Shanghai, China).

### 3.2. Process for Preparation of Msalais Wine and Their Procurement

The simulations took place in in the laboratory using the processing technology of DaoLang Musalais Co., Ltd., Awati, China. During the preparation of the Msalais samples, grape juice (GJ) was obtained by squeezing fresh grapes through a screw press. The extracted juice was transferred into a steam-boiling tank to boil for 12 h followed by being cooled in cold water to prepare condensed juice (CJ). Then, CJ was pumped in a stainless steel tank to be fermented for 45 days at conditions of fixation to prepare Msalais wine (MW). The aging of the MW was carried out for 12 months thereafter to acquire aged Msalais wine (AW). Five liters of each sample was packed in PET bottles and stored at −20 °C for further analysis.

### 3.3. LC-MS/QTOF Analysis

Initially, Msalais wine (MW) was centrifuged at 3000× *g* for 15 min at 4 °C. After that, the supernatant was filtered through a 0.22 μm PTFE membrane filter for analysis.

LC-MS/QTOF analysis was performed on an AB Triple TOF 5600 plus (AB SCIEX, Framingham, MA, USA) mass spectrometer equipped with Waters UPLC (Waters Corp., Milford, MA, USA) and a UV-vis detector. In this process, MS analysis was performed according to the procedure described by Pu [16], whereas slight modifications were made in the UPLC analysis.

A C18 type column (Agilent ZORBAX, 100 × 4.6 mm, i.d., 1.8 µm) was used to inject 5 μL of the sample at 25 °C, and the flow rate was adjusted to 0.8 mL/min. The mobile phases consisted of 0.1% formic acid (eluent A) and 0.1% formic acid acetonitrile (eluent B). The gradient program was as follows: 0 min, 95% A; 2 min, 95% A; 25 min, 50% A; 35 min, 5% A; 37 min, 5% A; 40 min, 95% A.

### 3.4. HPLC Analysis

The grape juice (GJ), condensed juice (CJ), Msalais wine (MW), and aged Msalais wine (AW) were centrifuged at 3000× *g* for 15 min at 4 °C. The supernatant was filtered through a 0.45 μm PTFE membrane filter for chromatographic analysis.

HPLC analysis was carried out using Waters 2695 HPLC coupled to a Waters 2489 UV–Vis detector (Waters Corp., Milford, MA, USA). A C18 type column (Agilent ZORBAX SB, 250 × 4.6 mm, i.e., 5 µm) was used at 25 °C for the separation of compounds. The mobile phases consisted of 0.1% formic acid solution (eluent A), methanol (eluent B), and acetonitrile (eluent C). A 10 μL sample was injected, and the flow rate was adjusted to 1 mL/min. The gradient program was as follows: 0–6 min: 85–59% A, 10–36% B; 6–30 min: 59–54% A, 36–44% B. The detection wavelengths were 280 nm and 360 nm for the real-time monitoring of the peak intensity.

### 3.5. DPPH Radical Scavenging Activity

The DPPH assay was performed according to the method adopted by Olivares-Tenorio [52]. Briefly, a working solution was prepared by dissolving DPPH (6 × 10^−5^ M) into 100% methanol, and then, 3.9 mL of this solution was mixed with 0.1 mL of the sample solution. The mixture was incubated in a water bath at 25 °C for 30 min in dark conditions. The change in the absorbance of the sample extract was measured at 515 nm for 30 min until the absorbance reached a steady state. A standard curve was prepared using different concentrations of Trolox. The results were expressed as μmol Trolox/L.

### 3.6. ABTS Radical Scavenging Activity

The ABTS assay was performed according to the method adopted by Koley [52]. Briefly, an aliquot (0.1 mL) of the sample solution was mixed with 3.9 mL of ABTS + solution. The mixture was maintained for 10 min in a dark place. The absorbance was measured at 30 min intervals at 714 nm. Trolex solution was used to prepare the standard curve, and the results were expressed as μmol Trolox/L.

### 3.7. Cupric-Reducing Antioxidant Capacity (CUPRAC)

The cupric-ion-reducing antioxidant capacity was determined according to the method by Koley [53]. Briefly, an aliquot (0.1 mL) of the sample solution was added in a mixture of solutions including 1 mL of CuSO_4_ (5 mM), 1 mL of neocuproine solution (3.75 mM), 1 mL of ammonium acetate buffer solution (1 mM), and 1 mL of water. Then, the mixture was mixed well and maintained for 30 min in the dark. The change in the absorbance of the sample extract was measured at 450 nm for 30 min until the absorbance reached a steady state. The standard curve was prepared using different concentrations of Trolox. The results were expressed as μmol Trolox/L.

### 3.8. Statistical Analysis

All measurements were conducted in triplicate, and data are presented in the form of mean and standard error. Statistical analysis was performed using the SPSS software (Version: 20.0, Armonk, NY, USA). The data were subjected to one-way analysis of variance (ANOVA), and the significant differences were evaluated using Duncan’s test (*p* < 0.05).

## 4. Conclusions

Msalais wine is a popular traditional drink with strong ethnic characteristics in China, but there is very limited information regarding its phytochemical composition and antioxidant activity, and this makes it very difficult to understand MW and assess the health benefits of consuming it. In this study, the chemical composition of MW was identified and characterized for the first time. A total of thirty-eight compounds, including eight furans, eleven phenolic acids, fourteen flavonoids, and five others, were identified via LC-MS/QTOF in the negative ion mode. Additionally, the effects of the winemaking stages on the major phenolic compounds and antioxidant activity were evaluated, and the results showed that condensation juice (CJ) had the highest levels of phenolic compounds and antioxidant activity because of its concentration in nature, while the levels were significantly decreased within the fermentation stage due to the hydrolysis and oxidation of polyphenols, which gradually stabilized during aging. These results indicate that the concentration and spontaneous fermentation stages are the key stages which affect the phenolic compounds and antioxidant activities in Msalais wine. Therefore, key microorganisms, the optimization of concentration technology, reductions in the fermentation period, and related environmental factors will be closely studied in the future. Moreover, grape skin and seeds should be added during the concentration and fermentation stages; thus, they can be regarded as good sources of potential phenolic compounds.

## Figures and Tables

**Figure 1 molecules-28-01250-f001:**
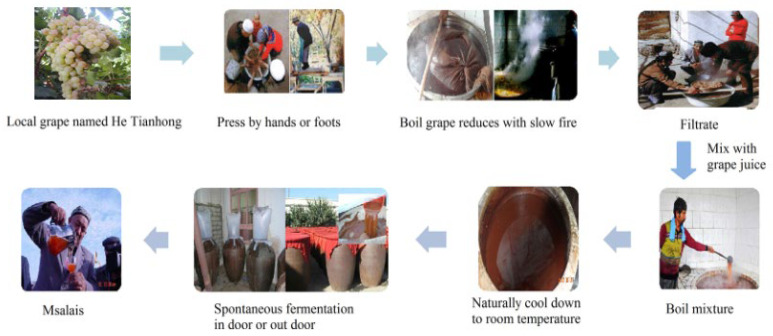
Traditional winemaking processes of Msalais.

**Figure 2 molecules-28-01250-f002:**
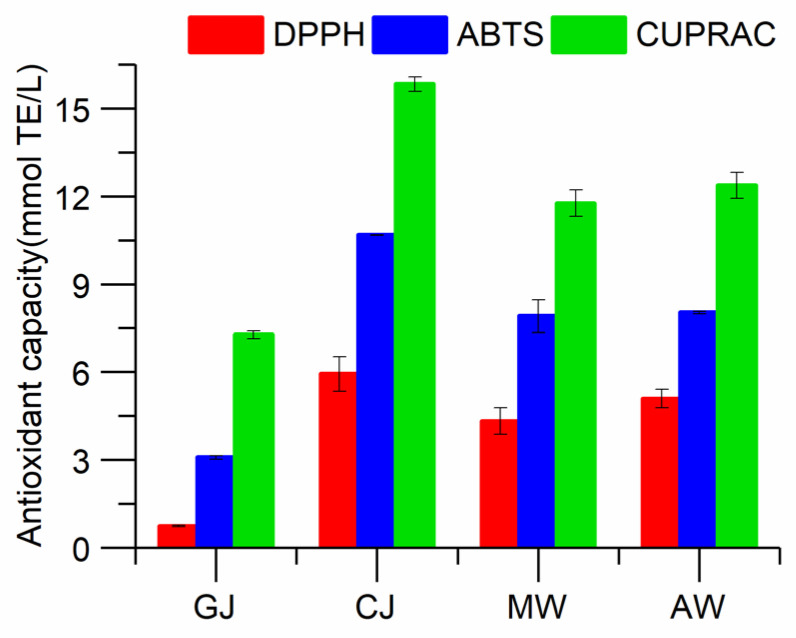
Effect of winemaking on DPPH, ABTS, and CUPRAC values.

**Table 1 molecules-28-01250-t001:** Compounds tentatively identified in Msalais wine via LC-MS/QTOF.

Peak	RT (min)	Tentative Identification	Formula	Error (ppm)	MM	[M-H]-	MS/MS Fragments
1	1.831	(3,4-dihydroxy-5-oxotetrahydro-2-furanyl)(hydroxy)-acetic acid	C_6_H_8_O_7_	1.4	192.0265	191.2000	191/111/87/85/57
2	2.167	L-erytho-pent-1-enofuranose	C_5_H_8_O_5_	6.8	148.0366	147.0309	147/129/87/85/57
3	2.930	garlic acid *	C_7_H_6_O_5_	2.1	170.0210	169.0146	169/125/124/79
4	3.732	3-hydroxy-2-butanyl 6-O-[3,4-dihydroxy-4-(hydroxymethyl)tetrahydro-2-furanyl]-glucopyranoside	C_15_H_28_O_11_	5.2	384.1626	383.1539	383/251/191/161/101/73
5	4.182	5-(hydroxymethyl)-2-furaldehyde	C_6_H_6_O_3_	2.1	126.0311	127.0387	109/81/53
6	4.611	3-hydroxyl-5-[1,2,3,4-tetrahydroxybutyl]-2(H)-furanone	C_8_H_12_O_7_	2.9	220.0583	219.0504	219/154/111/73
7	4.820	[4-(methoxycarbonyl)-5-methyl-2furyl]methyl-2,3-dyhydro-1,4-benzodioxine-2-carboxylate	C_17_H_16_O_7_	4.1	332.0896	331.0832	331/303/187/143/99
8	5.024	2-(3,4-dihydroxyphenyl)ethyl-3-glucopyranosyl- glucopyranoside	C_20_H_30_O_13_	4.5	478.1686	477.1592	477/323/153/123
9	5.202	3,5-dimethoxyphenyl-glucopyranoside	C_14_H_20_O_8_	1.4	316.1153	315.1081	315/153/123/122
10	5.372	protocatechuic acid *	C_7_H_6_O_4_	3.1	154.0261	153.0198	153/123/109/81
11	5.661	gentistic acid	C_8_H_10_O_3_	4.5	154.0630	153.0564	153/109
12	6.174	caftaric acid	C_13_H_12_O_9_	4.0	312.0476	311.0396	179/149/135/87
13	6.825	2-isopropyl-5-methylphenyl 6-O-[3,4-dihydroxy-4-(hydroxymethyl)tetrahydro-2-furanyl]-glucopyranoside	C_21_H_32_O_10_	3.8	444.1995	443.1906	443/189/119//89/71
14	7.085	p-hydroxybenzoic acid *	C_7_H_6_O_3_	0.9	138.1207	137.0258	137/91/79
15	7.175	1-(5-hydroxy-1-benzofuran-3-yl)ethanone	C_7_H_12_O_5_	4.6	176.0473	175.0620	175/157/115/85
16	7.346	Catechin *	C_15_H_14_O_6_	1.4	290.0743	289.0712	289/245/205/203/187/151
17	7.711	coutaric acid	C_13_H_12_O_8_	1.2	296.0527	295.0456	163/119/87
18	8.030	3-hydroxy-5-methylphenyl-6-O-glucopyranosyl- glucopyranoside	C_19_H_28_O_12_	3.8	448.1575	447.1491	447/401/269/161
19	8.315	ferulic acid *	C_10_H_10_O_4_	1.4	194.0574	193.0511	193/178/134/133
20	8.562	caffeic acid *	C_9_H_8_O_4_	5.7	180.0417	179.0359	179/149/135/134/87
21	9.120	Epicatechin *	C_15_H_14_O_6_	1.4	290.0743	289.0714	289/245/205/203/187/151
22	9.317	eriodictyol-7-O-glucopyranoside	C_21_H_22_O_11_	2.5	450.1157	449.1078	449/287/259/178/151
23	9.622	2,6-dimethoxy-4-[3-oxo-1-buten-1-yl]phenyl-glucopyranoside	C_18_H_24_O_9_	3.7	384.1421	383.1382	206/188/160/118
24	10.080	eriocitrin	C_27_H_32_O_15_	3.3	596.1741	595.1754	595/301/300/271/255
25	10.289	p-hydroxycinnamic acid *	C_9_H_8_O_3_	6.3	164.0468	163.0413	119/117/93
26	10.785	cinnamic acid	C_9_H_8_O_2_	4.4	148.0519	147.0458	119/103/76
27	11.376	Rutin *	C_27_H_30_O_16_	2.5	610.1528	609.1453	609/301/300/271/255/151
28	11.609	ellagic acid *	C_14_H_6_O_8_	2.0	302.0057	300.9984	300/283/244/229/201/145
29	11.762	quercetin-3-O-glucoside *	C_21_H_20_O_12_	3.7	464.0949	463.0865	463/301/300/271/255/179
30	12.182	quercetin 3-O-glucuronide	C_21_H_18_O_13_	3.1	478.0742	477.0660	477/301/255/179/151
31	12.804	kaempferol-3-O-rutinoside	C_27_H_30_O_15_	1.8	594.1579	593.1511	593/285/284/255/227
32	12.910	ramnazin-3-O-rutinoside	C_28_H_32_O_16_	3.7	624.1085	623.1068	623/315/314/300/299/271
33	13.067	quercetin-3-O-rhamnoside	C_21_H_20_O_11_	2.0	448.3769	447.0908	447/301/300/271/255/179
34	13.490	Resveratrol *	C_14_H_12_O_3_	0.7	228.0781	227.0712	227/185/157/143
35	14.085	Phloridzin *	C_21_H_24_O_10_	3.6	436.1224	435.1276	435/273/179/167/93
36	15.965	kaempferol-7-O-rhamnose	C_21_H_20_O_10_	3.9	432.0984	431.1013	431/285/284/255/227/185
37	20.016	Quercetin *	C_15_H_10_O_7_	2.6	302.0354	301.0349	301/273/179/151/121/93
38	21.465	Kaempferol *	C_15_H_10_O_6_	2.3	286.1102	286.1043	285/241/229/187/145

* Confirmed by authentic standards.

**Table 2 molecules-28-01250-t002:** Effects of winemaking on phenolic compounds (mg/L).

Sample Name	GJ	CJ	MW	AW
gallic acid	1.21 ± 0.07 ^d^	26.48 ± 0.56 ^c^	38.01 ± 0.07 ^b^	41.60 ± 1.13 ^a^
p-coumaric acid	6.43 ± 0.21 ^d^	13.03 ± 0.21 ^a^	10.43 ± 0.39 ^b^	7.38 ± 0.13 ^c^
(+)-catechin	89.65 ± 1.76 ^d^	272.92 ± 6.51 ^a^	201.03 ± 0.21 ^b^	175.94 ± 6.65 ^c^
(−)-epicatechin	6.86 ± 0.13 ^d^	42.63 ± 0.92 ^a^	37.13 ± 0.92 ^b^	28.62 ± 0.85 ^c^
caffeic acid	1.95 ± 0.35 ^c^	18.31 ± 0.07 ^b^	21.97 ± 0.21 ^a^	21.70 ± 0.7 ^a^
ferulic acid	2.16 ± 0.12 ^a^	1.91 ± 0.07 ^b^	0.90 ± 0.15 ^c^	1.01 ± 0.07 ^c^
rutin	6.96 ± 0.04 ^d^	10.68 ± 0.06 ^c^	11.36 ± 0.17 ^b^	11.81 ± 0.12 ^a^
resveratrol	0.99 ± 0.05 ^d^	7.81 ± 0.07 ^a^	4.78 ± 0.06 ^c^	6.33 ± 0.02 ^b^
quercetin	ND	1.98 ± 0.06 ^b^	3.89 ± 0.06 ^a^	3.60 ± 0.08 ^a^

^a^ The results are present as mean ± SD (*n* = 3), and values in each columns with different letters are significantly different (*p* < 0.05).

**Table 3 molecules-28-01250-t003:** Values of Pearson’s correlation coefficients (r).

	CUPRAC	ABTS	DPPH
quercetin	0.519	0.619	0.718
resveratrol	0.978 *	0.979 *	0.990 *
rutin	0.749	0.818	0.899
ferulic acid	−0.238	−0.352	−0.481
caffeic acid	0.757	0.830	0.897
epicatechin	0.932	0.966 *	0.935
catechin	0.978 *	0.978 *	0.918
p-coumaric acid	0.848	0.840	0.720
gallic acid	0.636	0.718	0.820

* Correlation is significant at the 0.05 level (2-tailed).

## Data Availability

Not applicable.

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
