# Peer review of "Effect of Winemaking on Phenolic Compounds and Antioxidant Activities of Msalais Wine"

_molecules, 2023, doi:10.3390/molecules28031250_

Round 1

Reviewer 1 Report

Authors present the assessment of wine making on phenolic compounds and antioxidant activities of Msalais wine. The manuscript is appropriately written, and it contains relevant contributions. However, my recommendation is to accept publication after the following minor revisions:

English and typo should be revised.

It would be interesting to add references of methods that are without it (for example, process for preparation of Msalais wine and chemical analysis).

It would be interesting to add if analyses of variance assumptions were guaranteed (homogeneity of variance, independent residuals and normal distribution of residuals).

It would be interesting to improve discussions about results from Table 2. Why have the concentrations of some compounds, like gallic acid and rutin, increased by aging?

It would be interesting to improve discussions about results from antioxidant activities. It would be interesting to compare these results with other ones from literature and to explain the mechanisms that led the decrease of antioxidant activity by fermentation. Why are some results without error bars in Figure 1? It would be interesting to improve discussions about results from Table 3. It would be interesting to discuss how these results are correlated to antioxidant activity.

Reviewer 2 Report

The idea is not clear and the research methodology is not clear, therefore I will reject this work.

1.   The authors provide a less than compelling abstract, and the focused findings are barely presented.

2.   For me, the introduction was poorly written and more of a statement of common sense with no more useful information in sight.   It needs to be added and improved.

3.   you should have a literature review section instead of jumping straight to methodology.   without a detailed literature review, I don't think this is a blank study.

4.   Lack of research methodology or detailed flow chart of the research process, without this, it is difficult to interest the reader and make clear the research done by the author.

5.   Little discussion of the results.   An in-depth comparison of your study with other similar studies will not only improve the quality of your manuscript, but also help to gain the attention of other authors.

6.   The language must be improved;   many sentences do not flow logically from one another and are difficult to read.   It is recommended to rewrite and then re-submit the manuscript.

Round 2

Reviewer 2 Report

The authors addressed most of my comments, while the quality of this manuscript has been greatly improved. I recommend accepting this manuscript after minor revisions. There are two small suggestions for the authors.

(1) The serial number before "Introduction" should be 1.

(2) The flow chart in Figure 2 may not make sense and needs to be carefully proofread.

Author Response

Dear editor:

Once again, we thank you for your professional review of our articles. As you were concerned, there were several issues that needed to be addressed. Based on your good suggestions, we have made extensive corrections to our previous draft, as detailed below.

We look forward to hearing from you for a favorable decision. Thank you again for your time and consideration.

Point 1: The serial number before "Introduction" should be 1. 

Response 1: Based on your comments, we have changed the serial number in front of "Introduction" to 1.

Point 2: The flow chart in Figure 2 may not make sense and needs to be carefully proofread.

Response 2: We think the comments you made are very correct. We have removed Figure 2.